# New Potentiality of Bioactive Substances: Regulating the NLRP3 Inflammasome in Autoimmune Diseases

**DOI:** 10.3390/nu15214584

**Published:** 2023-10-28

**Authors:** Baixi Chen, Yuhua Wang, Guangjie Chen

**Affiliations:** Department of Immunology and Microbiology, Shanghai Jiao Tong University School of Medicine, Shanghai 200025, China; beauregard_spencer@sjtu.edu.cn (B.C.); supperoliver@sjtu.edu.cn (Y.W.)

**Keywords:** NLRP3 inflammasome, rheumatoid arthritis, systemic lupus erythematosus, systemic sclerosis, bioactive substance

## Abstract

The NOD-like receptor family pyrin domain-containing 3 (NLRP3) inflammasome is an essential component of the human innate immune system, and is closely associated with adaptive immunity. In most cases, the activation of the NLRP3 inflammasome requires priming and activating, which are influenced by various ion flux signals and regulated by various enzymes. Aberrant functions of intracellular NLRP3 inflammasomes promote the occurrence and development of autoimmune diseases, with the majority of studies currently focused on rheumatoid arthritis, systemic lupus erythematosus and systemic sclerosis. In recent years, a number of bioactive substances have shown new potentiality for regulating the NLRP3 inflammasome in autoimmune diseases. This review provides a concise overview of the composition, functions, and regulation of the NLRP3 inflammasome. Additionally, we focus on the newly discovered bioactive substances for regulating the NLRP3 inflammasome in autoimmune diseases in the past three years.

## 1. Introduction

Autoimmune diseases are characterized by the abnormal activation of the immune system, leading to an assault on the body. The pathophysiology of this illness is complex, and the clinical manifestations demonstrate a broad spectrum of variability [1]. Rheumatoid arthritis, systemic lupus erythematosus, and systemic sclerosis are three representative autoimmune diseases.

The NOD-like receptor family pyrin domain-containing 3 (NLRP3) inflammasome serves as a pattern recognition receptor within cellular systems, responsible for detecting and responding to both danger- and pathogen-associated molecular patterns (DAMPs and PAMPs) [2]. Upon stimulation, the NLRP3 inflammasome is primarily activated through the classical pathway, leading to pyroptosis and the release of the pro-inflammatory cytokines IL-1β and IL-18. This process has wide-ranging biological effects, impacting both innate and adaptive immunity, and is closely associated with autoimmune diseases.

In the past decade, the effectiveness of targeting the NLRP3 inflammasome for treating diseases has been tested in various models, holding vast prospects for development [3]. Particularly in the last three years, a large number of bioactive substances regulating the NLRP3 inflammasome for treating autoimmune diseases have been gradually discovered, and their therapeutic effects have been tested in various animal models. Hence, the practice of promptly summarizing and synthesizing information proves advantageous in guiding future clinical experiments with medications of similar nature.

## 2. NLRP3 Inflammasome

### 2.1. Structure of the NLRP3 Inflammasome

The primary constituents of the NLRP3 inflammasome include NLRP3, apoptosis-associated speck-like protein (ASC), and caspase-1. NLRP3 is an intracellular sensor in the NLRP3 inflammasome, which is a pattern recognition receptor (PRR) present within the cell responsible for recognizing the DAMPs and PAMPs among the NLRP family members [4].The NLRP3 protein comprises three distinct structural domains: the N-terminal pyrin domain (PYD), the central nucleotide-binding oligomerization domain (NACHT or NOD), and the C-terminal leucine-rich repeat (LRR) domain containing 12 leucine residues. The structure of each portion is intricately interconnected with its respective function. The PYD is associated with the recruitment of ASC [4]; the NACHT possesses the capability to bind nucleotides and hydrolyze ATP, and its activation can result in PYD interactions, thereby recruiting ASC [5]; and LRR serves as an additional molecular switch in NLRP3 activation, interacting with the NIMA-related kinase 7 (NEK7) located at the microtubule organizing center [6]. This interaction opens the NLRP3 cage formed on the trans-Golgi network (TGN) before stimulation by molecules like nigericin, enabling the formation of active NLRP3 [7].

Beyond PYD, ASC contains a caspase recruitment domain (CARD), which interacts with NLRP3 and procaspase-1 individually [5]. When given no stimulation, the NLRP3 protein forms oligomers and becomes inactive. This occurs through the C-terminal of the LRR domain, which prevents its binding with ASC [7]. Upon receiving appropriate signals, NLRP3 assembles with ASC and procaspase-1 to form the NLRP3 inflammasome [8]. Caspase-1, functioning as an effector molecule, has the ability to cleave and activate gasdermin D (GSDMD) responsible for pyroptosis, simultaneously generating mature pro-inflammatory cytokines, which is crucial for regulating immune responses [9,10,11]. Figure 1 provides a concise representation of the structure of the NLRP3 inflammasome.

### 2.2. Activation of the NLRP3 Inflammasome

Currently, the categorization of NLRP3 inflammasome activation is generally classified into three types: canonical, non-canonical, and alternative pathways. In this review, we focus on the canonical pathway with a brief discussion of non-canonical and alternative pathways.

#### 2.2.1. Canonical Pathway

In the canonical pathway, the NLRP3 inflammasome necessitates the presence of both a priming signal and an activation signal [12]. The priming signal encompasses the regulation of NLRP3 expression and post-translational modifications (PTMs), although PTMs still maintain NLRP3 in a self-inhibited state [9,13]. The activation signal typically relies on changes in intracellular ion concentrations such as potassium ($K^{+}$) and calcium (${Ca}^{2+}$).

The priming of NLRP3 is commonly dependent on the NF-κB signaling pathway. Surface receptors, including PRRs and TNF receptor (TNFR) superfamily members, respond to various stimuli primarily by activating the ubiquitin-dependent kinase TAK1 and then the multi-subunit IκB kinase (IKK). Subsequently, the IKK complex mediates the phosphorylation and subsequent degradation of IκBα, resulting in the transient nuclear translocation of NF-κB heterodimers [12]. In the context of research on rheumatoid arthritis (RA), it has been observed that lipopolysaccharide (LPS) has a role in the regulation of synovial fibroblast pyroptosis by activating the NF-κB signaling pathway [14]. Other transcription factors such as Sp17, c-Myb, AP-2, and c-Ets [2], together with sterol regulatory element binding transcription factor 3 [15], contribute to the expression of NLRP3. This indicates that the activation signals of the NLRP3 inflammasome exhibit a wide range of variations, and these signals have the capacity to mutually influence one another [16].

Following priming, NLRP3 remains in an inhibited state and requires the involvement of an activation signal. Once activated, NLRP3 assembles together with ASC and caspase-1 to form an inflammasome, resulting in the complete activation of the NLRP3 inflammasome. According to current research, the process of activation is known to transpire through multiple pathways, with the $K^{+}$ efflux pathway and mitochondrial dysfunction being the ones most thoroughly investigated [4].

$K^{+}$ efflux is considered a fundamental mechanism for the ATP-induced activation of the NLRP3 inflammasome. V P E’trilli et al. found that elevated extracellular $K^{+}$ levels inhibit NLRP3 activation in human monocytes, while intracellular $K^{+}$ reduction triggers NLRP3 inflammasome activation [17]. Furthermore, classical activators of the NLRP3 inflammasome, such as nigericin and ATP, lead to intracellular $K^{+}$ reduction, and $K^{+}$ channel blockers inhibit the release of IL-1β [18], providing further evidence that the efflux of $K^{+}$ promotes the activation of the NLRP3 inflammasome. Mechanistically, purinergic receptor P2X purinoceptor 7 (P2X7), two-pore $K^{+}$ channel THIK-1, and two-pore domain $K^{+}$ channel ($K_{2P}$) TWIK2 have all been found to mediate $K^{+}$ efflux, subsequently activating the NLRP3 inflammasome [19,20]. In a model of PBMC-derived macrophages established to study RA, anti-citrullinated protein antibodies (ACPA) were found to activate pannexin channels, leading to ATP secretion and the subsequent activation of P2X7 receptors to promote $K^{+}$ efflux. This study first proved that ACPA can activate the NLRP3 inflammasome involved in the pathogenesis of RA [21]. NEK7 functions in the downstream pathway following $K^{+}$ efflux in the signal cascade of activation, mediating the assembly and activation of the NLRP3 inflammasome [22]. The latest research has demonstrated a notable rise in the development of the NEK7-NLRP3 complex in the context of Streptococcus pneumoniae infection, with an accompanying rapid phosphorylation of anaplastic lymphoma kinase and c-JunN-terminal kinase [23].

Additionally, ${Na}^{+}$ influx can promote the activation of NLRP3 induced by certain agonists [24]. However, it should be noted that the only influx of pure ${Na}^{+}$ is inadequate to activate NLRP3 [25]. A plausible explanation for this phenomenon is that the presence of ${Na}^{+}$ leads to the stimulation of $K^{+}$ efflux. Alternatively, it is also possible that ${Na}^{+}$ influx and $K^{+}$ efflux are regulated by a shared mechanism [26]. Chloride (${Cl}^{-}$) efflux is another regulatory pathway for activating NLRP3. During the process of activation, the intracellular chloride intracellular channel (CLIC) functions in a downstream manner following the $K^{+}$ efflux and mitochondrial reactive oxygen species (mtROS) axis. The translocation of CLICs to the plasma membrane is induced by mtROS, which is then followed by ${Cl}^{-}$ efflux. This process facilitates the interaction between NEK7 and NLRP3 [4]. It is evident that NEK7 acts downstream of both $K^{+}$ efflux in the signal cascade and ${Cl}^{-}$ efflux, hence aiding the formation of inflammasomes. Furthermore, ${Cl}^{-}$ efflux can also promote inflammasome assembly by inducing ASC oligomerization. Similar to ${Na}^{+}$, this facilitative action likewise depends on $K^{+}$ efflux [27].

However, not all cellular activations of NLRP3 rely on $K^{+}$ efflux. For instance, certain imidazoquinoline amines such as imiquimod and CL097, as well as peptidoglycans, have the ability to activate NLRP3 in a manner that is not dependent on $K^{+}$ efflux [28,29]. Hence, there exist NLRP3 activation mechanisms that bypass $K^{+}$ efflux in certain cells.

Dysfunctional mitochondria release mtROS and mitochondrial DNA (mtDNA), contributing to the activation of the NLRP3 inflammasome [30]. Research indicates that mtROS is primarily generated through electron transport chain (ETC) complexes I and III, and is associated with NLRP3 activation [31]. Nevertheless, the precise role of mtROS in the activation of NLRP3 is still not well understood. In more specific studies, ETC complexes I or III were replaced with alternative oxidases maintaining normal ETC function but incapable of producing mtROS. In the absence of mtROS, cells exhibited normal NLRP3 activation and IL-1β release [32], suggesting that mtROS may not be essential for NLRP3 inflammasome activation. Cytidine monophosphate kinase 2 is an enzyme that provides deoxyribonucleotides for mtDNA synthesis. It induces CAMK1-dependent IRF2 transcription through Myd88 and TRIF TLR signaling, and CAMK2-mediated mtDNA synthesis is necessary for oxidative mtDNA generation induced by NLRP3 agonists [33]. The release of mtDNA from mitochondria that have been damaged can occur within the cytoplasm, hence aiding in the activation of NLRP3. For example, oxidized mtDNA can bind and activate NLRP3 [34]. Moreover, it is essential to note that TLR-mediated mtDNA synthesis plays a critical role in the activation of NLRP3, as genetic mtDNA depletion impairs NLRP3 activation [33]. However, NLRP3 activation may also lead to mtDNA release [30]. Studies suggest that ETC participates in NLRP3 activation through the production of phosphocreatine-dependent ATP [32]. Therefore, mtDNA depletion may indirectly impact NLRP3 activation through its influence on ETC activity. Overall, the significance of mtDNA in NLRP3 activation is still being debated.

${Ca}^{2+}$, as an indispensable second messenger in many physiological processes, plays a crucial role in the activation of the NLRP3 inflammasome. The presence of extracellular ATP has the potential to facilitate ${Ca}^{2+}$ influx. The physiological functions of calcium-binding protein S100A9 are mediated by its interaction with ${Ca}^{2+}$ and can serve as an inflammation biomarker reflecting the severity of RA. A study revealed that IL-6 has the potential to facilitate the interaction between recombinant cathepsin B (CTSB) and NLRP3 via the CTSB/ATP pathway, thereby initiating the activation of the inflammasome and subsequently inducing RA. This mechanism was also validated in J774A.1 cells stimulated by IL-6 and ATP, resulting in an observed upregulation of S100A9 expression [35]. In another study investigating the pathogenesis of RA, it was discovered that colloidal calciprotein particles activate the calcium sensing receptor (CaSR) and induce the activation of phospholipase C through G proteins, leading to the hydrolysis of inositol triphosphate (IP3), among other effects. IP3 causes the release of calcium from intracellular calcium stores, leading to an elevation in intracellular calcium levels, perhaps facilitating the initiation of the NLRP3 inflammasome [36]. Acid-sensing ion channels (ASICs) represent cation channels activated by extracellular acidosis. Among them, ASIC1a possesses the distinct ability to facilitate the transportation of calcium ions. ASIC1a induces apoptosis of chondrocytes in RA by increasing intracellular ${Ca}^{2+}$ mediated by extracellular acidosis [37]. Additionally, excessive or sustained ${Ca}^{2+}$ influx can damage mitochondria and release mitochondrial mtROS, further activating the NLRP3 inflammasome [30,38].

Hypoxia-inducible factor-1α (HIF-1α) has also been found to be an inducer of NLRP3 inflammasome activation. It can be activated by various upstream signals, including ROS and the NF-κB signaling pathway [39]. In macrophages, the activation of HIF-1α is known to stimulate the synthesis of inflammatory factors, while inhibiting NF-κB can prevent excessive inflammation activation [40].

Finally, lysosomes and the TGN are also involved in the activation of the NLRP3 inflammasome. The engulfment of crystals by lysosomes leads to lysosomal damage and rupture, thereby activating the NLRP3 inflammasome. Self-dsDNA together with its autoantibodies can trigger the activation of the NLRP3 inflammasome in systemic lupus erythematosus (SLE) [41]. This activation process is potentially facilitated by the generation of ROS and the aforementioned $K^{+}$ efflux [7]. Recent research has identified that internalized C4b-binding protein (C4BP) acts as an inhibitor of the crystal- or particle-induced inflammasome response in primary human macrophages. In vivo, it has been observed that C4BP in mice has the ability to inhibit the augmentation of the inflammatory state [42].

The TGN is crucial for both $K^{+}$ efflux-dependent and independent activation of the NLRP3 inflammasome. $K^{+}$ efflux has been demonstrated to be necessary for the recruitment of NLRP3 to the TGN [43], possibly by reducing cellular ionic strength to facilitate ion binding. Additionally, studies have indicated that protein kinase IKKβ plays a critical role in recruiting NLRP3 to dispersed TGN (dTGN) [44]. NLRP3 agonists trigger the formation of dTGN, which then undergoes transportation to the microtubule-organizing center (MTOC). Through the formation of ionic bonds between the negatively charged phosphatidylinositol-4-phosphate and the positively charged polybasic region between the PYD and NACHT domains of NLRP3, NLRP3 is recruited to dTGN [43]. Subsequently, NLRP3 associates with NEK7 in the centrosome, resulting in the formation of active NLRP3 inflammasome specks. Following this, NLRP3 recruits ASC through PYD–PYD interactions, and ASC recruits caspase-1 through CARD–CARD interactions, ultimately forming the NLRP3 inflammasome [45]. Figure 2 presents a comprehensive depiction of the principal signaling pathways implicated in the classical activation of the NLRP3 inflammasome.

#### 2.2.2. Other Pathways

In addition to the canonical pathway, there exist non-canonical and alternative pathways, which will be briefly outlined here.

The non-canonical inflammasome refers to the caspase-11-dependent inflammasome in mice (or caspase-4 and caspase-5 in humans), where Nur77 functions as an intracellular LPS receptor and activates the non-canonical NLRP3 inflammasome [46,47]. This leads to pyroptosis triggered by GSDMD cleavage [48]. During various bacterial infections, cytoplasmic LPS directly binds to mouse caspase-11 (or human caspase-4/caspase-5), leading to its oligomerization and activation [49]. Caspase-11 plays a crucial role in septic shock [50]. In human and porcine monocytes, alternative inflammasome activation is mediated by the upstream TLR1-TRIF-RIPK8-FADD-CASP4 signaling cascade. In contrast to canonical inflammasomes, the alternative response does not elicit the generation of pyroptotic entities or pyroptosis [51].

### 2.3. Inducing Pyroptosis

After the activation of the NLRP3 inflammasome, self-cleavage and activation of caspase-1 are induced. The activation of caspase-1 leads to the maturation of the pro-inflammatory cytokines IL-1β and IL-18. Simultaneously, it cleaves GSDMD and releases its N-terminal domain. This domain translocate to the cell membrane, forming pores that mediate the release of cellular contents and induce inflammatory cell death, known as pyroptosis [52]. The clearance of infected cells and resistance to pathogens are facilitated by this mechanism, wherein the activation of the NLRP3 inflammasome plays a critical role in the host’s defense against pathogenic invasions and the maintenance of homeostasis.

Nevertheless, an overabundance of NLRP3 inflammasome activation can contribute to the advancement of diverse inflammatory disorders, including the autoimmune diseases referenced in the following subsection.

### 2.4. Regulation of the NLRP3 Inflammasome

The NLRP3 inflammasome exhibits both a precise mechanism of activation and a dependence on meticulous regulation. The application of stringent regulatory mechanisms guarantees that the activation of the NLRP3 inflammasome elicits a targeted immune response that is both effective in providing protection and devoid of any detrimental effects on the organism. When regulatory mechanisms get disrupted, it can potentially result in the development of several autoimmune diseases. Numerous studies have demonstrated that the development of several autoimmune disorders is linked to the disruption of NLRP3 inflammasome activation [53]. Based on different PTMs regulating the NLRP3 inflammasome, the most common regulatory mechanisms currently studied involve phosphorylation and ubiquitination. The regulators of these two mechanisms act on different sites on NLRP3. We have compiled and listed them in Table 1 and Table 2.

## 3. The Biological Functions of NLRP3 Inflammasome

The principal effectors of the biological response of the NLRP3 inflammasome are IL-1β and IL-18, which are generated through the cleavage of caspase-1. By affecting different immune cells and tissue cells, they play an essential role in immune response.

As a classical inflammatory mediator, IL-1β is primarily secreted by monocytes, macrophages and dendritic cells. Dendritic cells are of paramount importance in the process of recognizing antigens by naïve T cells and initiating specific immune responses. The interaction between IL-1β and IL-1R plays an essential part in the process of dendritic cell maturation [72]. IL-1β can promote the functionality of various T cell subsets. For example, it stimulates the secretion of IL-17 by CD4+T cells and γδT cells [73], facilitates the infiltration of Th17 cells [74], and promotes the proliferation and functional activation of CD8+ T cells [75]. The aforementioned impacts contribute positively to the host’s capacity to resist specific pathogens [76]. IL-1β also mediates the cross-talk between innate and adaptive immunity [77].

IL-18 is a cytokine that is also classified under the IL-1 family. Nevertheless, the expression of IL-1β is more prevalent in comparison [78]. IL-18 binding protein (IL-18BP) is a natural inhibitor of IL-18, which aids in regulating the function of IL-18. IL-12 performs a pivotal role in modulating the biological function of IL-18. In the presence of IL-12, IL-18 induces Th1-type responses and stimulates the production of IFN-γ. Conversely, IL-18 alone contributes to the induction of Th2-type responses. IL-18 is additionally involved in the stimulation of NK cells, hence contributing to the immune responses mounted against both infectious agents and tumors [79]. Therefore, IL-18 exhibits pleiotropic biological functions in the body and primarily participates in immune responses through T cells and NK cells. Several inflammatory diseases and autoimmune diseases are easily treated more manageably by decreasing the level of IL-18 in the body [80,81].

In summary, the NLRP3 inflammasome-mediated release of IL-1β and IL-18 simultaneously affects innate immunity and adaptive immunity. Dysfunctional NLRP3 inflammasomes are involved in the development of various autoimmune diseases.

## 4. NLRP3 Inflammasome in Autoimmune Diseases

### 4.1. Rheumatoid Arthritis

RA is a typical representative of the autoimmune disease. Its typical clinical manifestations include symmetrical swelling and pain in multiple joints, as well as involvement of extra-articular organs. Chronic synovial inflammation serves as the principal pathological alteration in RA [82]. Although the exact pathogenesis of RA has not been fully elucidated, multiple immune cells and cytokines are considered important drivers [83]. IL-1β and IL-18, generated upon NLRP3 inflammasome activation, emerge as critical pro-inflammatory mediators in innate immunity, exerting a pivotal influence on the initiation and advancement of RA [84].

The collagen-induced arthritis (CIA) mouse model is the most commonly used animal model for RA [85]. The model demonstrates an apparent augmentation in the expression of NLRP3 inside synovial tissues, exhibiting a favorable correlation with both clinical and radiographic assessments of arthritis [86]. The examination of peripheral blood cells in patients with RA indicates that individuals in the active phase of the disease have elevated baseline levels of NLRP3 inflammasomes [87]. Among them, monocytes exhibit a marked increase in the expression and activation of the NLRP3 inflammasome [88], whereas neutrophils demonstrate a notable decrease in NLRP3 inflammasome expression [89]. In spite of the activation of NLRP3 inflammasomes, there is no notable disparity in the peripheral concentrations of IL-1β in RA patients [88]. However, it may be elevated at specific infiltrated sites, including the lungs of RA patients with interstitial pneumonia [90] and synovial tissues [91]. Compared to IL-1β, it is plausible that IL-18 exhibits a stronger correlation with the potential joint involvement [92].

The main contributors responsible for the establishment and maintenance of a persistent and prolonged inflammatory milieu within the joints affected by RA are the infiltration of macrophages and the activation of fibroblasts. The infiltration of macrophages is positively correlated with the extent of joint erosion, which is considered an early sign of RA [93]. In RA patients, the activation level of the NLRP3 inflammasome in macrophages within synovial tissue is significantly increased [94], and finally, the pyroptosis can impact the biological characteristics of fibroblast-like synoviocytes (FLS) [95], demonstrating the bridging role of the NLRP3 inflammasome in connecting immune cells and non-immune cells in RA. ACPA serves as a significant biomarker within the population of RA patients. Previous studies have shown that ACPA has the capacity to trigger the activation of the P2X7 ion channel, consequently driving the activation of the NLRP3 inflammasome in macrophages of individuals diagnosed with RA. Consequently, this activation leads to the release of IL-1β and aggravates the symptoms and progression of RA [21]. Furthermore, Mir-155 functions as a critical regulatory factor of NLRP3 and has been found to regulate NLRP3-mediated cell pyroptosis in various diseases, including osteoarthritis. However, its effects vary across different cell types. Gen Li et al. [96] reported its inhibitory effect on chondrocyte pyroptosis, while Chen Li et al. [97] reported its promotion of macrophage pyroptosis. Mir-155 levels are elevated in RA patients, facilitating the polarization of M1 macrophages [98] and contributing to the activation of CD4+ T cells [99]. Cytokines secreted by M1 macrophages, such as IL-6, can further activate NLRP3 via the cathepsin B/S100A9 pathway, promoting the progression of RA [35].

FLS are the primary cellular constituents of the synovium, playing a crucial role in the initiation and progression of inflammation. The synovial tissue of RA patients expresses high levels of TLR2/4, providing a solid foundation for the priming of the NLRP3 inflammasome [100]. Multiple substances have been proven to activate the NLRP3 inflammasome within RA-FLS, including the metabolic product succinate [101], TNF-α [102], LPS [103], and phospholipase C-like 1 [104]. In addition, the synovial tissue of RA patients is known to be in a chronic hypoxic state [105], which facilitates the sustained activation of HIF-1α-related signaling pathways in FLS [106] and pyroptosis mediated by NLRP3 [107]. MiR-223 [108] and SMAD [95] have been shown to inhibit the activation of NLRP3 in RA-FLS, with the latter exerting its effects through the TGF-β pathway. Pyroptosis of RA-FLS may result in abnormal proliferation and migration, as well as the release of a large amount of inflammatory cytokines and chemokines [109]. IL-18 induces angiogenesis in RA synovial tissue, contributing to vascularization [110]. And due to high expression levels of IL-1R in the synovial tissue of RA patients [111], FLS exposed to IL-1β exhibit an enhanced secretion of inflammatory mediators, including IL-6, IL-8, and matrix metalloproteinases. This heightened release of inflammatory cytokines contributes to the degradation of connective tissues within the joints. Activated fibroblasts can also secrete chemokines and growth factors to recruit monocytes and promote their proliferation [112]. Therefore, there is a reciprocal interaction between FLS and monocytes/macrophages mediated by the NLRP3 inflammasome [113].

Moreover, the pyroptosis of chondrocytes should not be overlooked. One of the fundamental characteristics of RA is the presence of an acidic environment inside the joint cavity. In mouse models of arthritis, the presence of extracellular acidosis triggers the activation of NLRP3-mediated pyroptosis in chondrocytes. This activation may occur via two mechanisms: firstly, ASICs facilitate calcium influx [114], and secondly, it can be induced by an elevation in ROS levels [115]. Consequently, the aforementioned process culminates in the occurrence of pyroptosis in chondrocytes, thus indicating the potential role of NLRP3-mediated chondrocyte pyroptosis in the pathogenesis of cartilage tissue degradation. The inhibitory effects of Mir-144 on this pathway have been shown [116]. Figure 3 provides a comprehensive overview of the involvement of NLRP3 inflammasomes in the pathophysiology of RA.

### 4.2. Systemic Lupus Erythematosus

SLE is an autoimmune disease characterized by immune dysregulation in patients, manifested by the formation of autoantibodies, deposition of immune complexes, and multi-organ damage, with the skin and kidneys being particularly affected [117]. Increasing evidence suggests that the NLRP3 inflammasome plays a crucial role in the pathogenesis of SLE, especially in relation to the abnormal activation of the innate and adaptive immune systems [118]. The abnormal activation of the NLRP3 inflammasome has been detected in various cell types within SLE patients, including macrophages [119], monocytes [120], renal tubular epithelial cells [121], and podocytes [122]. Additionally, several substances present in SLE patients contribute to the activation of the NLRP3 inflammasome within different cell types, including self-derived dsDNA [41], anti-dsDNA antibodies [123], and neutrophil extracellular traps (NETs) [124].

In addition, IL-1β and IL-18 in the blood of SLE patients also show significant alterations. Different studies have reported varying levels of IL-1β in SLE patients [125,126], while the levels of IL-18 are elevated and closely associated with active renal involvement [125]. At the organ level, the activation of the NLRP3 inflammasome can also be detected, primarily in the kidneys [127] and skin [128]. NLRP3 inflammasome activation often correlates with the severity of clinical indicators in patients, most commonly manifested as proteinuria due to renal involvement [129].

The pathogenesis of SLE involves the critical participation of neutrophils and IFN-γ in the innate immune system, with the NLRP3 inflammasome likely promoting SLE through them [130]. Upon activation, neutrophils release NETs [131], which contain diverse enzymes and antimicrobial peptides, and potentially contribute to autoimmunity [132]. Altered levels of NETs in SLE patients impact neutrophil bactericidal functions and facilitate the development of autoimmunity and fibrosis [133]. A decade ago, J Michelle Kahlenberg et al. discovered that NLRP3 activation in the macrophages of SLE patients contributes to the release of NETs mediated by neutrophils and impairs their clearance, resulting in NET accumulation [124]. This process triggers caspase-1 activation, initiating a detrimental cycle of inflammation [134] and potentially leading to severe complications [135]. IFN-I, an essential immunoregulatory factor, aids in the activation of adaptive immune cells and mediates immune dysregulation in SLE patients [136]. IFN-related genes are associated with genetic susceptibility to SLE, and elevated serum IFN levels enhance disease activity and increase the risk of relapse in SLE patients [137], thereby promoting the dysfunction of endothelial progenitor cells [138]. The interplay between IFN and the NLRP3 inflammasome has been gradually elucidated, including the induction of NLRP3 inflammasome activation by IFN-I during influenza A virus infection [139]. In SLE patients, IFN-I activates inflammasomes in monocytes through IRF-1 mediation [140]. Certain environmental pollutants, such as bisphenol A, also activate the NLRP3 inflammasome via IFN-I signaling [141]. Bisphenol A was previously identified to upregulate the expression of Nur77 [142], and recent studies have revealed that Nur77 serves as a critical mediator of NLRP3 activation through noncanonical pathways [143], potentially offering a novel perspective on the underlying mechanism of NLRP3 activation induced by bisphenol A. However, the activated NLRP3 inflammasome seems to negatively regulate IFN-I signaling, which may be associated with the SOCS1-mediated regulation of IRF-3 [144].

Another key characteristic of SLE in the adaptive immune response is the generation of autoantibodies. Autoantibodies against self-dsDNA directly bind to TLR or activate the NLRP3 inflammasome through promoting ROS or K+ efflux [145], thereby driving the excessive activation of B cells [146]. Recently, the NLRP3 inflammasome in Tfh cells has been found to involve B cell activation, leading to the production of high-affinity antibodies and correlating with disease activity [147].

Although the NLRP3 inflammasome plays a critical role in driving the development of SLE through various mechanisms, the relationship between the two is not straightforward. There is still controversy regarding the relationship between the activation level of NLRP3 and SLE. Furini et al. indicated that the activation level of NLRP3 inflammasomes and the expression level of P2X7 receptors are decreased in macrophages in SLE patients. It is suggested that IL-6 may play a more important role compared to IL-1β [148]. Qing-rui Yang et al. put forward the notion of a negative association between the activation level of the NLRP3 inflammasome in peripheral blood mononuclear cells (PBMCs) derived from patients with SLE and the severity of the disease, highlighting the involvement of IFN-I in this relationship [149]. Zhen-zhen Ma et al. also observed lower expression levels of NLRP3-related molecules in the serum of SLE patients [150].

Furthermore, the absence of NLRP3 can exacerbate the severity of SLE in certain circumstances. The direct blockade of IL-1β alone does not effectively reduce renal inflammation and overall survival rates in the NZM2328 mice, which even exacerbates the severity of proteinuria [151]. NLRP3/ASC promotes renal glomerular damage by increasing T cell infiltration [152], but mice lacking NLRP3 or ASC also develop severe lupus nephritis, which may be associated with other signaling pathways such as TGF-β [153]. This suggests that the NLRP3 inflammasome plays distinct roles in different pathways in SLE patients. Therefore, investigating how to selectively inhibit detrimental NLRP3 signaling holds potential significance for the treatment of SLE.

### 4.3. Systemic Sclerosis

Systemic sclerosis (SSc) is an autoimmune disease characterized by vascular abnormalities, inflammatory responses, and fibrosis. The pathogenesis of early-stage SSc primarily involves endothelial dysfunction, while in later stages, fibroblast hyperactivity leads to excessive extracellular matrix secretion, resulting in fibrosis in multiple organs. Inappropriate inflammatory responses are one of the core features of SSc, which ultimately facilitates irreversible organ damage [154]. Upon activation of the NLRP3 inflammasome, the secretion of IL-1β and IL-18 can promote vascular abnormalities, inflammatory responses, and extracellular matrix secretion [155]. Additionally, GSDMD-N-mediated pyroptosis has also been implicated in SSc [156].

Given the clinical characteristics of SSc, there are currently several models that demonstrate the typical features of fibrosis in SSc, including TSK gene mutation mice, Fra-2 transgenic mice, Fli-1 gene knockout mice, and bleomycin-induced fibrosis mice [157]. Among them, bleomycin-induced fibrosis mice are the most commonly used model. Bleomycin activates TLR2, mediates the release of various inflammatory cytokines, and ultimately leads to pulmonary fibrosis in mice [158]. In mice treated with bleomycin, the levels of caspase-1, IL-1β, and IL-18 are significantly increased in serum and lung tissue. This fibrosis can be alleviated by knocking out the caspase-1 and IL-18 genes [159]. The degree of change in NLRP3-related molecules in SSc patients varies in different studies. For example, some studies have indicated a significant increase in the content of caspase-1 in the skin tissue of SSc patients [160], while the level of caspase-1 in peripheral blood is significantly decreased [161]. In addition, Emily Lin et al. found a lack of significant alteration in the level of IL-1β in the serum of SSc patients [162], while Y.-J. Zhanget al. showed a significant increase [163]. The expression levels of NLRP3, IL-1β, and IL-18 in the skin of SSc patients are significantly elevated [160]. The results of these different studies may suggest that the activation of NLRP3 in different tissues of SSc patients varies and is related to the specific disease progression, such as higher expression levels of IL-1β and caspase-1 in the muscle tissue of SSc patients with myositis [164]. Furthermore, the levels of NLRP3-related cytokines in SSc patients are closely correlated with certain clinical indicators. High levels of IL-18 in the serum indicate poor pulmonary function in patients, while high levels of IL-1β in skin tissue indicate severe skin fibrosis [160] and relatively better lung function [162]. However, there are also studies that refute the clinical predictive value of IL-1β levels in the serum [163].

Mechanistically, the NLRP3 inflammasome may drive the pathogenesis of SSc by affecting monocytes/macrophages, B cells, endothelial cells, and fibroblasts. Monocytes circulate in the blood and serve as precursors for tissue macrophages. Studies have shown that monocytes from SSc patients produce more ROS [165], which contributes to the activation of the NLRP3 inflammasome. This may lead to the secretion of IL-1β and TNF-α [166]. Additionally, infection with parvovirus B19 can activate the NLRP3 inflammasome and enhance the activity of monocytes, resulting in increased release of IL-1β [167]. When monocytes migrate into tissues and differentiate into macrophages, they can also induce inflammation and promote fibrosis, highlighting the significant role of macrophages in the pathogenesis of SSc [168]. Macrophages can be classified into classically activated (M1) or alternatively activated (M2) macrophages based on their activation patterns. M2 macrophages secrete pro-fibrotic factors such as TGF-β, CCL18, and PDGF, exhibiting potential profibrotic functions [168]. An imbalance in the differentiation of M1/M2 macrophages is one of the mechanisms underlying SSc [169]. IRF8, a transcription factor promoting M1 macrophage differentiation, is reduced in SSc patients. Consequently, the proportion of M2 macrophages significantly increases, and artificially downregulating IRF8 expression worsens symptoms in SSc mice [170]. Recent studies have revealed the role of IRF8 in regulating the activation of the NLRP3 inflammasome, providing a new perspective on the association between NLRP3 inflammasome and fibrosis [171]. Upon sustained activation of the NLRP3 inflammasome, high levels of IL-1β in the bloodstream promote the differentiation of monocytes into macrophages through endothelial cell mediation. These macrophages then secrete high levels of pro-fibrotic factors such as CCL18, CCL2, and CXCL8, exacerbating skin fibrosis in SSc patients [172]. Furthermore, altered levels of macrophage migration inhibitory factor (MIF) have been observed in different tissues of localized and diffuse SSc patients [173]. MIF contributes to the interaction between NLRP3 and the intermediate filament protein vimentin, which facilitates NLRP3 inflammasome activation [174].

In addition to monocytes/macrophages, B cells are another group of immune cells that contribute to the pathogenesis of SSc. The dysregulation of B cell function in patients leads to their infiltration into affected organs, where they secrete antibodies and interact with other immune cells, promoting the fibrotic process in SSc [175]. B cell activating factor (BAFF) plays a pivotal role in bleomycin- and IL-17-mediated pulmonary fibrosis [176]. Subsequent studies have found that NLRP3 is one of the regulatory targets of BAFF. The upregulation of BAFF activates the NF-κB pathway, promoting the expression of NLRP3 while simultaneously activating ROS and K+ efflux, thereby activating the NLRP3 inflammasome in B cells [177]. However, this process can be inhibited by miR-30a [178]. Additionally, the secretion of IL-1β by activated B cells can also impact IgM synthesis [179].

The damage and dysfunction of endothelial cells in the vascular wall are evident throughout the entire process of SSc pathogenesis [180]. Endothelial dysfunction is typified by an aberration in the balance between vasodilation and vasoconstriction, accompanied by increasing ROS and inflammatory mediators [181], which are closely associated with the NLRP3 inflammasome. The release of IL-1β following activation can further induce the expression of endothelin-1 and adhesion molecules in endothelial cells, promoting leukocyte migration and maintaining vascular inflammation [181].

Fibrosis is the ultimate outcome of SSc, and myofibroblasts are key participants in this process. Epithelial–mesenchymal transition is one of the important features of fibrosis, characterized by the loss of polarity in epithelial cells and their transformation into mesenchymal cells. This is accompanied by an increased expression of α-SMA and a decreased expression of adhesion molecules, along with an increased deposition of extracellular matrix. The NLRP3 inflammasome promotes epithelial–mesenchymal transition [182]. Mitochondrial dysfunction is commonly present in fibrotic diseases, and the resulting mtROS is one of the important signals for activating NLRP3 molecules, which also has an effect on activating fibroblasts [183]. Upon IL-1β signaling through IL-1R on fibroblasts, it may form a positive feedback loop, ultimately resulting in increased collagen synthesis through the complex interplay of cytokines and cellular interactions [184]. In addition to fibroblasts, the NLRP3 inflammasome is also involved in lung fibrosis mediated by alveolar epithelial cells. The activation of the NLRP3 inflammasome in alveolar epithelial cells promotes the differentiation of mesenchymal stem cells into myofibroblasts during the pulmonary fibrosis process [185]. Figure 4 summarizes the role of the NLRP3 inflammasome in the pathogenesis of SSc.

## 5. Bioactive Substances Regulating the NLRP3 Inflammasome for the Treatment of Autoimmune Diseases

Considerable advancements have been achieved in the realm of pharmaceutical research pertaining to the targeting of the NLRP3 inflammasome as a therapeutic approach for the management of autoimmune disorders, which holds important practical significance and potential applications. Bioactive substances constitute a diverse family, encompassing active components extracted from plants as well as certain physiological substances that are essential for maintaining organismal life. Compared to chemically synthesized drugs, most of them exert milder effects [186]. Although the specific target sites and exact functions of many bioactive substances are not yet fully elucidated, it has been discovered in recent years that some of them indeed exhibit significant regulatory effects on NLRP3, including inhibiting the activation of the NLRP3 inflammasome or regulating NLRP3-associated signaling pathways. This review provides a concise overview of the bioactive substances that have been investigated for their potential to regulate the NLRP3 inflammasome in RA, SLE, and SSc within the past three years.

### 5.1. Inhibiting the Activation of the NLRP3 Inflammasome

The vitamin D receptor (VDR), expressed in nearly all immune cells, mediates the biological effects of vitamin D and regulates immune responses associated with autoimmune diseases [187]. VDR directly binds to NLRP3 and inhibits NLRP3 inflammasome activation by suppressing BRCC3-mediated ubiquitination. Furthermore, the inhibitory effect of VDR is enhanced by vitamin D [188]. Subsequent investigations have provided evidence indicating that vitamin D inhibits the activation of the NLRP3 inflammasome via a SIRT3-SOD2-mtROS signal [189]. These findings partially explain the protective effect of vitamin D in autoimmune diseases, including its ability to reduce the levels of inflammatory cytokines and ROS in RA patients [190]. Additionally, vitamin D improves renal damage in lupus mouse models by inhibiting the NF-κB and MAPK pathways and reducing autoantibody levels [191].

Compound K, a metabolite of ginsenoside derived from ginseng, inhibits NLRP3 inflammasome activation by suppressing NF-κB signaling and reducing ROS levels. This inhibition leads to the suppression of M1 macrophage activation and the induction of M2 macrophage activation [192]. Compound K also promotes SIRT1-mediated autophagy, which is one of the mechanisms by which it inhibits NLRP3 activation [193]. Notably, it demonstrates significant therapeutic effects in lupus nephritis (LN) [194].

Myrtenal and β-caryophyllene oxide are active components extracted from Myrica rubra. They alleviate symptoms in mice with antigen-induced arthritis (AIA) by inhibiting NLRP3 inflammasome activation [195].

### 5.2. Regulating the TLR4/NLRP3/NF-KB/GSDMD Signaling Pathway

Several bioactive substances with unidentified specific targets have been found to exert their effects by regulating the TLR4/NLRP3/NF-κB/GSDMD signaling pathway. These include traditional Chinese medicine formulas such as the Jinwujiangu capsule for treating RA [196], Baihu-Guizhi decoction [197], and Qi-Sai-Er-Sang-Dang-Song decoction [198]. Additionally, bioactive compounds extracted from plants, such as the monomer derivative of paeoniflorin [199], wedelolactone derived from Xanthium sibiricum [200], licorice-processed DGN products containing Huangruixiang [201], a combination of mangiferin and cinnamic acid [202], punicalagin [203], tectoridin [204], strychnine combined with Atractylodes Macrocephala [205], and pinocembrin from the genus Pinus [206], have also shown activity through this pathway.

### 5.3. Regulating Other Pathways Related to NLRP3

Honokiol is a bioactive biphenolic compound derived from plants. It has been shown to improve kidney function and reduce albuminuria in LN mice, exerting a preventive and therapeutic effect on LN. This may be achieved by inhibiting the NLRP3 inflammasome and sirtuin 1 signaling in the kidneys, thereby suppressing aberrant crosstalk between renal macrophages and tubular epithelial cells [207]. Subsequent studies have further elucidated the mechanism of the SIRT1-mediated negative regulation of NLRP3 inflammasome activation [208].

ASIC1a is an extracellular acid-activated cation channel that plays a role in chondrocyte pyroptosis by activating NLRP3 through calpain-2/calcineurin in the acidic microenvironment of extracellular acidosis [114]. Yiqi Yangyin Tongluo acts on this pathway to alleviate chondrocyte pyroptosis [209].

HIF-1α is an important mediator in the hypoxic synovial microenvironment of RA patients. It promotes the expression of the NLRP3 gene through the transcription factor STAT3. Monomeric derivatives of paeoniflorin extracted from paeoniflorin [107] and sodium Danshensu extracted from Salvia miltiorrhiza [210] alleviate the clinical manifestations of RA by regulating the HIF-1α/STAT3/NLRP3 pathway.

Er Miao San (EMS) is a formula consisting of equal proportions of Atractylodes Macrocephala rhizome and Cortex Phellodendri. It exhibits good anti-inflammatory effects in an animal model of RA, which is associated with the regulation of cytokine levels and the balance of Treg/Th17 cells [211]. Subsequent studies have found that EMS reduces macrophage polarization towards the M1 phenotype by regulating the miRNA-33/NLRP3 pathway [212], and serum containing EMS has a similar effect [213].

Table 3 provides a detailed summary of the relevant information on bioactive substances.

## 6. Conclusions

The activation of the NLRP3 inflammasome is notably disrupted in autoimmune illnesses such as RA, SLE, and SSc. IL-1β and IL-18 play a crucial role in facilitating the initiation of immune responses, serving as a vital link between innate and adaptive immunity. Although the results are not completely consistent, most of the studies have shown that the level of NLRP3-related proteins has a significant association with diverse clinical features observed in autoimmune diseases. Within certain microenvironments, the activation of the NLRP3 inflammasome and subsequent tissue degradation exhibit a mutually reinforcing relationship, hence establishing a positive feedback loop. This intricate process involves the active participation of various cellular components, including macrophages, lymphocytes, and tissue cells such as fibroblasts. Therefore, it is imperative to investigate the activation and regulatory mechanisms underlying NLRP3 and devise appropriate tailored pharmaceutical interventions. In recent years, an increasing number of biologically active substances have been discovered to possess regulatory functions on NLRP3 inflammasomes. While this is not their exclusive function, it opens up new perspectives for the treatment of autoimmune diseases and warrants further exploration.

## Figures and Tables

**Figure 1 nutrients-15-04584-f001:**
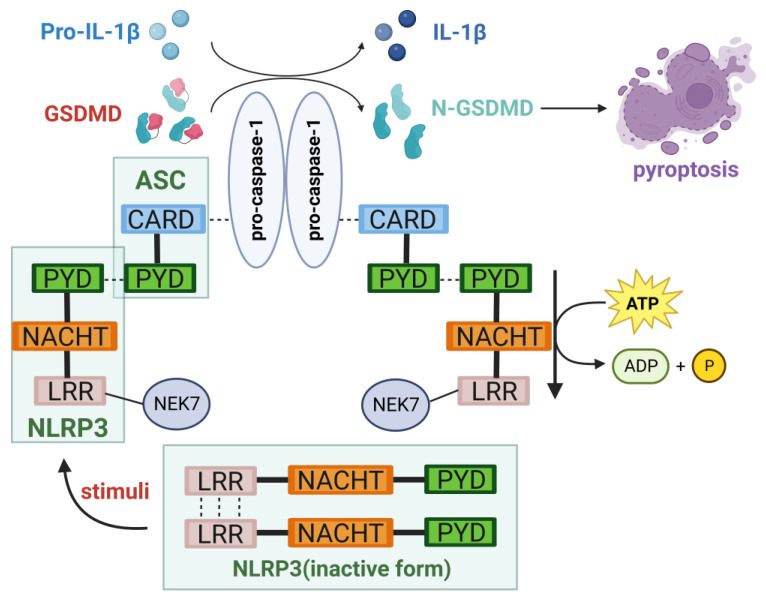
The structure of the NLRP3 inflammasome. The NLRP3 inflammasome is composed of NLRP3, ASC, and caspase-1. Upon activation, it mediates cellular pyroptosis, leading to the production of IL-1β and IL-18, exerting biological functions (created with BioRender.com, accessed on 21 October 2023).

**Figure 2 nutrients-15-04584-f002:**
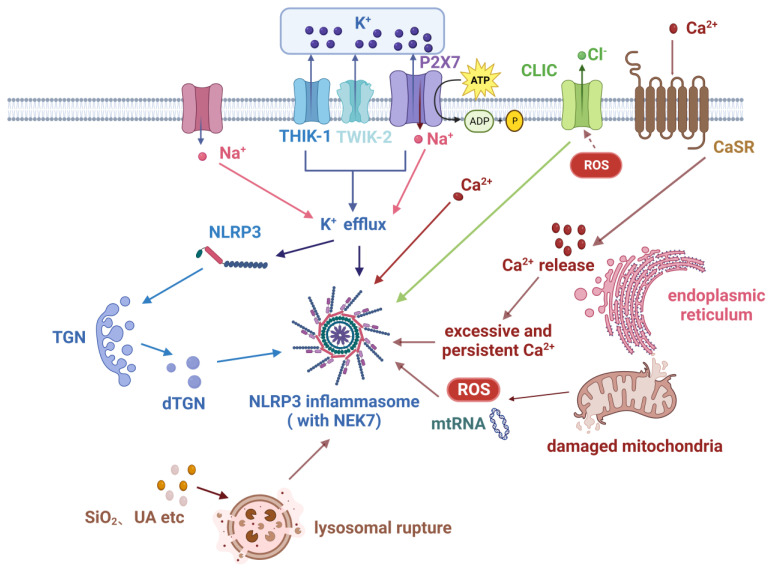
Key signaling pathways in the canonical activation of the NLRP3 inflammasome. The participation of numerous signals is necessary for the activation of the NLRP3 inflammasome, with $K^{+}$ efflux being the most pivotal factor. In addition, ${Na}^{+}$ influx and ${Cl}^{-}$ efflux are also synergistic signals. The endoplasmic reticulum, trans-Golgi network, mitochondria, and lysosomes may be involved in NLRP3 inflammasome activation partly through these signals (created with BioRender.com).

**Figure 3 nutrients-15-04584-f003:**
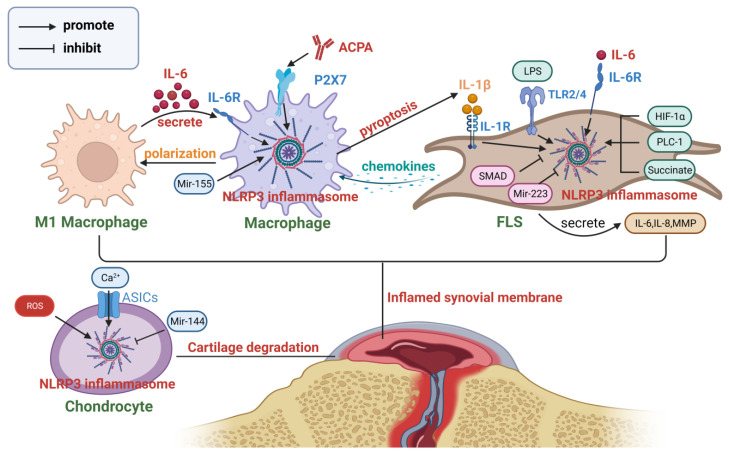
The role of the NLRP3 inflammasome in the pathogenesis of RA. Chondrocytes, macrophages, and FLS interact with each other through the NLRP3 inflammasome, collectively promoting the occurrence and progression of RA (created with BioRender.com).

**Figure 4 nutrients-15-04584-f004:**
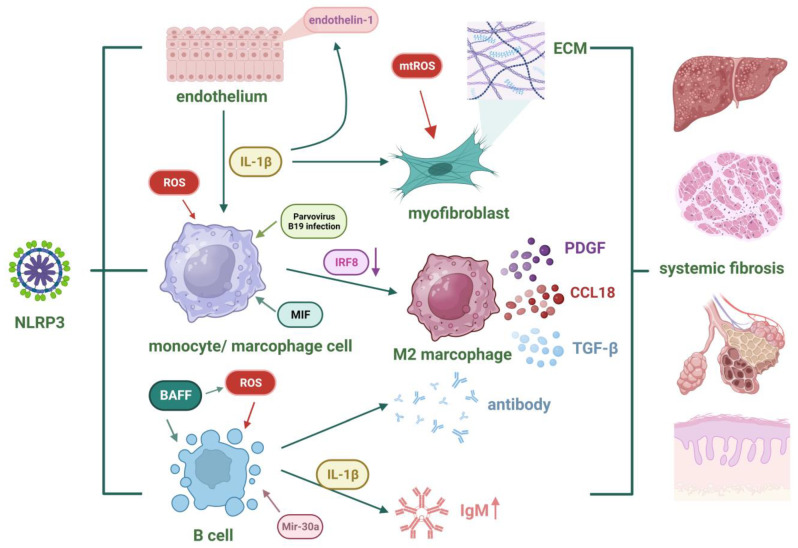
The role of the NLRP3 inflammasome in the pathogenesis of SSc. Fibrosis represents the ultimate outcome of SSc, and NLRP3 facilitates fibrosis through monocytes/macrophages, B cells, endothelial cells, and myofibroblasts (created with BioRender.com).

**Table 1 nutrients-15-04584-t001:** The regulation of NLRP3 through dephosphorylation and phosphorylation.

PTM	Effect on NLRP3 Activation	Enzyme	Site on NLRP3	Reference
Dephosphorylation	Promote	PP2A	Ser5 in PYD domain	[54]
		PTEN	Tyr32 in PYD domain	[55]
Phosphorylation	Inhibit	CSNK1A1	Ser803 in LRR domain (mouse)	[56]
		EphA2	Tyr136 (Tyr132 in mouse)	[57]
		Serine/threonine protein kinase AKT	Ser5 in PYD domain	[58]
	Promote	Stress-activated protein kinase JNK1	Ser198 (Ser194 in mouse)	[59]
		BTK	Tyr136, Tyr140, Tyr143, and Tyr168	[60]
		Pak1	Thr659	[61]
		PKD	Ser295 in NACHT domain	[62]

Abbreviations: PP2A: phosphatase 2A; PTEN: protein tyrosine phosphatase; EphA2: ephrin type-A receptor 2; BTK: Bruton’s tyrosine kinase; Pak1: p21-activated kinase 1; PKD: protein kinase D.

**Table 2 nutrients-15-04584-t002:** The regulation of NLRP3 through ubiquitination and deubiquitination.

PTM	Effect on NLRP3	Enzyme	Site on NLRP3	Reference
Ubiquitination	Promote	BRCC3	Unknown	[63]
Deubiquitination	Promote	USP7/USP47	Unknown	[64]
		HUWE1	Lys27	[65]
		Ubc13	Lys567, Lys 689 (Lys 565, Lys 687 in mouse)	[66]
	Inhibit	ARIH2	Unknown	[67]
		gp78	Unknown	[68]
		Cbl-b	Lys 496 (Lys 492 in mouse)	[69]
		β-TrCP1	Lys384 (Lys380 in mouse)	[70]
		Cullin1	Lys 689 (Lys687 in mouse)	[71]

Abbreviations: BRCC3: BRCA1/BRCA2-containing complex subunit 3; USP7/47: Ubiquitin-specific protease 7/47; HUWE1: HECT, UBA and WWE domain-containing E3 ubiquitin protein ligase 1; Ubc13: ubiquitin E2-conjugating enzyme 13; ARIH2: Ariadne homologue 1; gp: glycoprotein; Cbl-b: Casitas B lymphoma-b; β-TrCP1: β-transducin repeat-containing protein 1.

**Table 3 nutrients-15-04584-t003:** Bioactive substances regulating the NLRP3 inflammasome to treat autoimmune diseases.

Mechanism	Bioactive Components	Note	Disease	Model	Effects
Inhibiting the activation of the NLRP3 inflammasome	Vitamin D	-	RA	HEK293T cells [188] and RA patients	Reducing the levels of inflammatory cytokines and ROS in RA patients [190]
	VDR agonist/VD3 [214]	-	SLE	MRL/Lpr mice	Decreasing urine protein and serum anti-dsDNA antibody levels
	Compound K [193,194]	Major absorbable intestinal bacterial metabolite of ginsenosides	SLE	ASLN mice	Improving renal function, albuminuria, and renal lesions, and reducing serum levels of anti-dsDNA
	Myrtenal and β-caryophyllene oxide [195]	Screened from Liquidambaris Fructus	RA	AIA mice	Reducing IL-1β and TNF-α in serum; attenuating the upregulation of NLRP3 and IL-1β expression in the synovial tissue
	Sulforaphene [215]		RA	CIA mice; mouse synovial macrophages	Specifically binding to NLRP3 and inhibiting its activation in tissues; alleviating arthritis and suppressing M1 macrophage
Regulating the TLR4/NLRP3/NF-KB/GSDMD signaling pathway	Jinwujiangu capsule [196]	A traditional Chinese medicine	RA	RA-FLS	Decreasing the expression of caspase-1, GSDMD, NLRP3, and ASC, suppressing the expression of IL-1β and IL-18
	Baihu-Guizhi decoction [197]	A traditional Chinese medicine-originated disease-modifying anti-rheumatic drug prescription	RA	AA rats	Reducing levels of TLR4, NLRP3, IL-1β, and IL-18; attenuating the redness and swelling of joints, arthritis incidence, and diameter of the limb
	Qi-Sai-Er-Sang-Dang-Song Decoction [198]	A Tibetan classical herbal formula	RA	RA-FLS	Down-regulating the levels of NLRP3 and relative cytokines
	Monomer derivative of paeoniflorin [199]	Paeoniflorin could inhibit the development and progression of arthritis in experimental animal models of arthritis	RA	AA rats	Inhibiting macrophage polarization and pyroptosis; attenuating bone erosion, soft tissue swelling, and joint space narrowing
	Wedelolactone [200]	Derived from Eclipta alba	RA	CIA rats	Ameliorating ankle joint swelling and cartilage degradation; decreasing the release of pro-inflammatory cytokines
	Licorice-processed DGN products [201]	Daphnes Cortex is a popular traditional Chinese herbal medicine for traumatic injuries and RA	RA	CIA rats; LPS-induced RAW264.7 cells	Ameliorating RA symptoms; regulating inflammatory cytokines, matrix metalloproteinases, and vascular endothelial growth factor
	Combination of mangiferin and cinnamic acid [202]	May be active components of the Baihu-Guizhi decoction	RA	AA rats RAW264.7cells; MH7A cells	Ameliorating arthritis severity, suppressing NLRP3 inflammasome-induced pyroptosis
	Punicalagin [203]	Active substance extracted from pomegranate peel	RA	CIA rats	Alleviating the high expression of inflammatory cytokines in synovial tissue; shifting macrophages to the M2 phenotype
	Tectoridin [204]	Isolated from the dry rhizome of iris	RA	RA-FLS	Hindering cell proliferation; markedly promoting apoptosis rates
	Strychnine combined with Atractylodes Macrocephala [205]	-	RA	MH7A cells	Promoting the apoptosis of synovial cell; reducing the level of TLR4and NLRP3
	Celastrol [216]	Extracted from Tripterygium wilfordii	RA	AA rats; THP-1 cells	Decreasing the arthritis index score; ameliorating joint swelling and synovial hyperplasia
	Euphorbium total triterpenes [217]	Mainly characteristic constituents of euphorbium	RA	AA rats	Relieving swelling; decreasing inflammatory cytokines
	Pinocembrin [206]	A flavonoid with anti-inflammatory effects	SSc	RAW264.7 and J774A.1 cells; BLM induced mice	Relieving pulmonary inflammatory response
Regulating the SIRT1/autophagy/NLRP3 axis and inhibiting the NLRP3/IL-33/ST2 axis	Honokiol	A major anti-inflammatory bioactive compound in Magnolia officinalis	SLE	NZB/WF1mice; MRL/Lpr mice	Improving renal function, albuminuria, and renal pathology; regulating T cell functions and reducing anti-dsDNA autoantibodies in serum
Regulating the ASIC1a/NLRP3 signaling pathway	Yiqi Yangyin Tongluo	Raw astragalus 30 g, dendrobium 15 g, polygala 15 g, achyranthes bidentata 25 g, honeysuckle 15 g	RA	AA rats	Significantly reducing the infiltration of inflammatory cells in the soft tissues; relieving the swelling of the ankle joints
Regulating the HIF-1α/NLRP3 pathway	Monomeric derivatives of paeoniflorin	-	RA	RA-FLS	Decreasing levels of HIF-1α and GSDMD-N; inhibiting FLS pyroptosis
	Sodium Danshensu	A structurally representative water-soluble derivative of Danshen	RA	CIA mice	Reducing the serum levels of IL-1β and IL-6; ameliorating paw oedema and bone destruction
	Dihydroarteannuin [218]	Extracted from the traditional Chinese herb Artemisia annua L.	RA	CIA mice; THP-1 cells	Reducing the serum levels of IL-1β and IL-6; alleviating paw oedema and bone destruction
Regulating the GBP5/P2X7/NLRP3 pathway	Sinomenine	Isolated from Sinomenii Caulis	RA	CIA mice; RAW264.7 cells	Alleviating arthritis symptoms; reducing the levels of inflammatory cytokines
Regulating the miRNA-33/NLRP3 pathway	Er Miao San	A traditional Chinese medicine composed of Atractylodis Rhizoma and Phellodendri Cortex in a 1:1 weight ratio	RA	AA rats, RAW264.7 cells, MH7A cells	Decreasing the paw volume and polyarthritis index; alleviating ankle joint histopathology, regulating Th17/Treg and M1polarization

## Data Availability

Not Applicable.

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
