# Peer review of "New Potentiality of Bioactive Substances: Regulating the NLRP3 Inflammasome in Autoimmune Diseases"

_nutrients, 2023, doi:10.3390/nu15214584_

Round 1

Reviewer 1 Report

Comments and Suggestions for Authors

Thank you for compiling this review. I have a few minor suggestions for improvement. Please consider replacing review article citations with primary research manuscripts wherever feasible. While you can retain the review citations, it is essential to emphasize actual research papers as primary sources.

Regarding the figures, they could benefit from enhanced clarity. It might be challenging to discern where to begin and conclude. Additionally, please give more prominence to NLRP3 within the figures.

I would suggest a thorough revision, particularly focusing on refining references and language formatting. Moreover, improving the clarity of the figures could enhance understanding, reducing the need to refer extensively to the main text.

Comments on the Quality of English Language

The article requires additional attention regarding the use of fonts, such as spacing and the inclusion of extra dots.

Author Response

Reviewer1 #

Thank you very much for your comments and professional advice. As you are concerned, there are several problems that need to be addressed. According to your nice suggestions, we have made extensive corrections to our previous manuscript, the detailed corrections are listed below.

  1. “Please consider replacing review article citations with primary research manuscripts wherever feasible. While you can retain the review citations, it is essential to emphasize actual research papers as primary sources.”

The author’s reply:We gratefully appreciate for your valuable suggestion. Based on it, we have extensively read the review citations in this manuscript and made efforts to replace some of them with corresponding primary research. The specific references include 5, 74, 82, 83, 95, 97, 109, 110, 116, 135, 136, 142, 169, 172, and 173. The corresponding statements have been highlighted in bluefont within the text. However, due to certain statements in the article covering broader topics that are difficult to summarize with several primary research, we still retain a small portion of the review citations.

2.“Regarding the figures, they could benefit from enhanced clarity. It might be challenging to discern where to begin and conclude. Additionally, please give more prominence to NLRP3 within the figures.”

The author’s reply: We have comprehensively revised Figure 4 to enhance its clarity and emphasized the role of NLRP3 inflammasome,making it more visually apparent. In addition, we optimized the layout in Figure 3 to make NLRP3 inflammasome more prominent. In addition, we also coordinated the styles of all four graphics to improve visual consistency. The resolution of all figures has been increased to 600 dpi.

3.“I would suggest a thorough revision, particularly focusing on refining references and language formatting. Moreover, improving the clarity of the figures could enhance understanding, reducing the need to refer extensively to the main text.”

The author’s reply: We have provided detailed responses to the questions regarding references and figures in the previous two points. As for language formatting, we have made every effort to optimize the manuscript. These modifications do not affect the content and structure of the paper. We have highlighted in orange font in the revised manuscript. We sincerely appreciate the valuable suggestion of the reviewer and hope that our corrections will be approved.

We would like to thank the referee again for taking the time to review our manuscript, and hope that our corrections will be approved.

Yours sincerely,

Bai-xi Chen, Yu-hua Wang, Guangjie Chen

October 21, 2023

Reviewer 2 Report

Comments and Suggestions for Authors

The reviewer appreciates the authors' interesting review of NLRP3-mediated inflammasome and bioactive substances targeting it as a therapeutic approach in autoimmune diseases. The NLRP3 inflammasome and pyroptosis/cell death is a vast subject to be delineated in one review and the reviewer applauds the authors for great work. In the same context, we know that there are many reviews on this pathway, however, the information on bioactive substances targeting the NLRP3 inflammasome pathway is the attention grabber. Moreover, the author must understand that most of the bioactive substances included in the study show pleiotropic efficacies (w.r.t other signaling pathways) rather than targeting preferentially NLRP3 inflammasomes only.

Minor Comments:

# The reviewer would like to recommend the inclusion of more studies carried out on bioactive substances that preferentially targeted NLRP3 biomarkers. Since, the novel section in this review is about bioactive substances, the information/discussion should be made on bioactive substances and their relationship with NLRP3 inflammasome. 

Also, the author must rethink the word "targeted" in their headings/title as these bioactive substance acts on other pathways as demonstrated by earlier studies. 

#The author must recheck their manuscript again as there are many words without spacing. Some of it is at Lines # 49, 54,55, 168, 202, 246, 548

# There are no references added in the introduction section. Please add them accordingly. 

Author Response

Reviewer 2 #

Thank you very much for your comments and professional advice. As you are concerned, there are several problems that need to be addressed. According to your nice suggestions, we have made extensive corrections to our previous manuscript, the detailed corrections are listed below.

1.“The reviewer would like to recommend the inclusion of more studies carried out onbioactive substances that preferentially targeted NLRP3 biomarkers. Since, the novel section in this review is about bioactive substances, the information/discussion should be made on bioactive substances and their relationship with NLRP3 inflammasome.  ”

The author’s reply:We have spared no efforts to supplement the fifth part of the manuscriptby incorporating the latest relevant research and previously overlooked studies. This includes Sulforaphene, Celastrol, Euphorbium total triterpenes, Dihydroarte-annuin, and Sinomenine. The corresponding additions have been highlighted in red font in Table 3 of the manuscript. The revised manuscript essentially encompasses all relevant research conducted within the past three years. Additionally, we have provided a more detailed explanation of the relationship between bioactive substances and the NLRP3 inflammasome at the beginning of the fifth part of the manuscript and in the conclusion section. The corresponding additions have also been marked in red font.

  1. “Also, the author must rethink the word "targeted" in their headings/title as these bioactive substance acts on other pathways as demonstrated by earlier studies.”

The author’s reply: We are very sorry for our negligence of using the word “targeted”. After careful consideration, we believe that the term "regulate" better reflects the relationship between the mentioned bioactive substances and the NLRP3 inflammasome in this article. We have made modifications to the relevant statements in the manuscript and highlighted them with a yellow background font,including the title, subtitles, and body text.

3.“The author must recheck their manuscript again as there are many words without spacing. Some of it is at Lines # 49, 54,55, 168, 202, 246, 548”

The author’s reply:We sincerely apologize for our oversight. After rechecking the entire document, we have added spaces in the respective sections to ensure that the manuscript meets the formatting requirements.

4.“There are no references added in the introduction section. Please add them accordingly.”

The author’s reply:We thank the reviewer for pointing this out, and we have added the relevant citations in the introduction section.

We would like to thank the referee again for taking the time to review our manuscript, and hope that our corrections will be approved.

Yours sincerely,

Bai-xi Chen, Yu-hua Wang, Guangjie Chen

October 21, 2023